# DATA FILTERING NETWORKS

**Alex Fang**[*,1,2]    **Albin Madappally Jose** [1]    **Amit Jain**[1]
**Ludwig Schmidt**[2]    **Alexander Toshev**[1]    **Vaishaal Shankar**[1]
[1]Apple    [2]University of Washington

## ABSTRACT

Large training sets have become a cornerstone of machine learning and are the foundation for recent advances in language modeling and multimodal learning. While data curation for pre-training is often still ad-hoc, one common paradigm is to first collect a massive pool of data from the Web and then filter this candidate pool down to an actual training set via various heuristics. In this work, we study the problem of learning a *data filtering network* (DFN) for this second step of filtering a large uncurated dataset. Our key finding is that the quality of a network for filtering is distinct from its performance on downstream tasks: for instance, a model that performs well on ImageNet can yield worse training sets than a model with low ImageNet accuracy that is trained on a small amount of high-quality data. Based on our insights, we construct new data filtering networks that induce state-of-the-art image-text datasets. Specifically, our best performing dataset DFN-5B enables us to train state-of-the-art CLIP models for their compute budgets: among other improvements on a variety of tasks, a ViT-H trained on our dataset achieves 84.4% zero-shot transfer accuracy on ImageNet, out-performing models trained on other datasets such as LAION-2B, DataComp-1B, or OpenAI's WIT. In order to facilitate further research in dataset design, we also release a new 2 billion example dataset DFN-2B and show that high performance data filtering networks can be trained from scratch using only publicly available data.

## 1 INTRODUCTION

Carefully curated datasets have driven progress in machine learning for decades, from early pattern recognition experiments in Bell Labs to recent developments like GPT-4, Stable Diffusion, and CLIP (Highleyman & Kamentsky, 1959; LeCun et al., 1989; 1998; Deng et al., 2009; Krizhevsky et al., 2009; 2012; Radford et al., 2019; 2021; 2022; OpenAI, 2023). Despite their crucial role, datasets themselves are rarely the subject of active research (Sambasivan et al., 2021).

Current approaches to improving performance on machine learning tasks have focused on scaling model capacity or training data volume. While scaling laws (Hestness et al., 2017; Kaplan et al., 2020; Aghajanyan et al., 2023; Cherti et al., 2023) have elucidated the relationship between model size, data size, and performance, little formal guidance exists on how to scale these quantities. On the model side, experimentation is straightforward - with enough compute, permutations of width, depth, normalization and training hyperparameters can be rigorously evaluated, leading to consistent modeling improvements over the years (Touvron et al., 2023a;b; Elsen et al., 2023).

The dataset side is unfortunately murkier. Most large-scale training sets are not released, leaving the community to attempt open reproductions (Schuhmann et al., 2021; 2022; Byeon et al., 2022; Gao et al., 2020); however, these are often one-off efforts without the iterative refinement that models enjoy. Recent efforts like DataPerf, DataComp and MetaCLIP (Mazumder et al., 2022; Gadre et al., 2023; Xu et al., 2023) help bridge the gap by providing consistent dataset evaluation and reproduction frameworks.

We argue dataset design can leverage the same tools as model design. Almost all large-scale dataset construction can be broken down into two phases: uncurated data collection and dataset filtering. We focus our work on the latter, with the assumption that a large uncurated dataset exists. We show data filtering networks (DFNs) - neural networks designed to filter data - can induce massive, high-quality pre-training datasets. Unlike previous techniques relying on domain-specific heuristics,

---

[*]Work done while at Apple

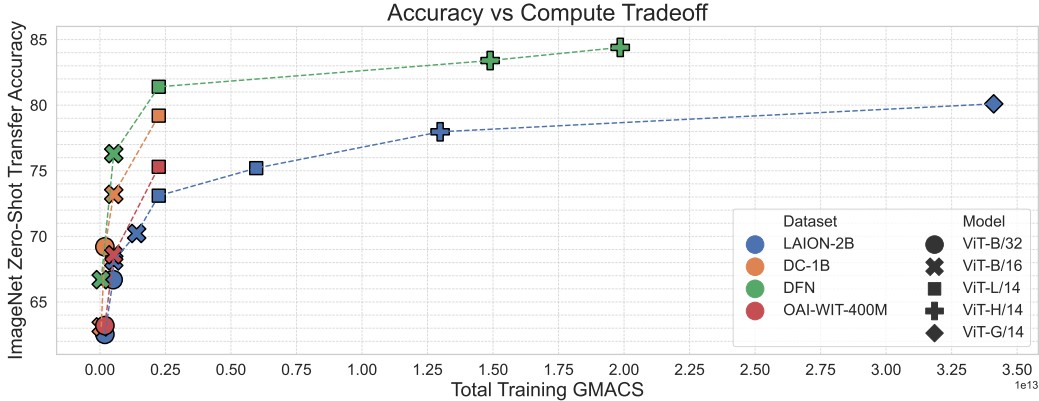

Figure 1: Compute scaling behavior of training CLIP models on various datasets. DFN-2B, the subset of CommonPool (DataComp-12.8B) chosen by our best performing data filtering networks, outperforms all other datasets including OpenAI's WIT and the previous state-of-the-art CLIP training dataset DataComp-1B. Our ViT-L outperforms a ViT-G trained on LAION with 18× more compute. Similarly, our ViT-B/16 outperforms OpenAI's ViT-L/14 trained with 4× more compute. Our ViT-H/14 achieves 84.4% on ImageNet, out-performing any model in its compute class. All DFN-trained models were trained on DFN-2B, except for the ViT-H which was trained on DFN-5B. Both datasets were induced by the same DFN. We note the cost of training DFN was omitted from this plot, which corresponds to less than $\frac{1}{50}$th of total CLIP training cost.

DFNs paired with a large unfiltered image-text pool produce billion-scale state-of-the-art datasets algorithmically. We demonstrate DFNs can be efficiently trained from scratch and improved with the same techniques as standard ML models.

The contributions of this work are as follows. First, we characterize the properties of data filtering networks that lead to high-quality datasets. We ablate properties of data filtering networks from supervision signal to training data quality. We find that a small contrastive image-text model trained on *only* high-quality data is sufficient to construct state-of-the-art datasets.

Second, we use these properties to train DFNs and construct datasets that induce Contrastive Image-Text Pre-trained (CLIP) models that achieve high accuracy and present better compute accuracy tradeoff than any existing dataset in the literature as show in Figure 1. In particular we train a ViT-L/14 or 12.8B examples seen on our DFN induced dataset DFN-2B to 81.4 ImageNet zero-shot transfer accuracy, outperforming the previous best ViT-L trained on DataComp-1B by over 2 percentage points. We further train a ViT-H/14 on a larger DFN induced dataset DFN-5B to 84.4 ImageNet zero-shot transfer accuracy. We show that models trained on these datasets show consistent improvements on many tasks, including zero-shot classification, retrieval, and visual question answering, and maintain the favorable robustness properties of CLIP models.

Lastly, the above insights can be used as a recipe to construct high-quality datasets from scratch by using only public data [1] thus making strides towards democratization of large high-quality datasets. In addition, we release DFN-2B for the community to enable research on large image-text models.

## 2 BACKGROUND AND RELATED WORK

### 2.1 CONTRASTIVE IMAGE LANGUAGE PRE-TRAINING (CLIP)

CLIP has altered the use of cheaply available image-alt-text datasets by demonstrating the practicality of large-scale training on web-scraped image-text pairs to build state-of-the-art image representations. CLIP consists of separate vision and text encoders, and uses contrastive loss during training to push the representations of related images and text pairs together, and unrelated pairs apart. Cru-

---

[1]Most large public image-text datasets including LAION-5B and DataComp-1B are built using OpenAI's CLIP model

cial to this process is a large dataset of *aligned* image-text pairs - images paired with semantically relevant text. The release of CLIP was followed by several other image-text models such as ALIGN, BASIC, LiT and Open-CLIP all of which we will refer to in this work as CLIP models (Jia et al., 2021; Pham et al., 2023; Zhai et al., 2022b; Ilharco et al., 2021). CLIP models generally come in 3 canonical sizes of vision transformer: ViT-B/32, ViT-B/16 and ViT-L/14; since then, the open source community has extended these to 3 larger variants ViT-H/14, ViT-g/14 and ViT-G/14 (Dosovitskiy et al., 2020; Zhai et al., 2022a). Generally the larger models exhibit better zero-shot generalization and transfer properties. CLIP models have been trained on a variety of datasets from OpenAI's WiT, Google's WebLI and JFT-3B, LAION, COYO and DataComp-1B.

Prior work has also studied how to fine-tune CLIP models to improve performance in targeted directions. CLIP models can be fine-tuned on image classification tasks by using templates to transform labels to text (Fang et al., 2022; Goyal et al., 2022). Additionally, practitioners often use weight ensembling to preserve robustness properties of the pre-trained model while reaping the benefits of fine-tuning (Wortsman et al., 2022). We take advantage of these techniques in order to improve the filtering models we train in this work.

## 2.2 DATASET CONSTRUCTION

Prior to CLIP, datasets most commonly used in computer vision were *supervised* with human labels (Deng et al., 2009; Krizhevsky et al., 2009). Though these older dataset construction pipelines were quite intricate and did not scale beyond a few million examples, they share some similarity with current constructions. Classical datasets such as ImageNet and CIFAR started with a large *roughly curated* pool of images paired with metadata, and used humans to either label or filter the data.

Modern dataset pipelines have a similar procedure but at a much higher scale. The initial pool of images can contain up to 100 billion images, and the dataset filtering is purely automated, often with a set of rules and heuristic filtering stages (Jia et al., 2021). Past work in natural language processing has used binary filters as an initial step to remove low quality documents (Wenzek et al., 2019; Brown et al., 2020), but contain multiple components to their filtering pipelines.

One of the first publicly available web-scale image-text datasets is LAION. LAION-400M and LAION-2B were constructed by collecting image-text pairs from Common Crawl, filtering by English, and keeping pairs whose image and text are well *aligned*. This alignment is performed using a procedure known as *CLIP filtering*, which uses an existing image-text model (in LAION's case OpenAI CLIP ViT-B/32), and removes samples whose cosine similarity between image and text are below some threshold. We show pseudocode of the basic CLIP filtering operation in Appendix H.

While CLIP filtering is convenient it is dependent on a existing trained CLIP model, and perhaps limited on the top-line performance of any model trained using it as a filter. For example, despite LAION-2B being five times larger than OpenAI's dataset, models trained on it could only match OpenAI's ImageNet zero-shot performance with a significantly larger compute budget.

To better facilitate the study of image-text datasets, researchers created the DataComp benchmark (Gadre et al., 2023). The benchmark provides 12.8 billion image-text pairs from Common Crawl so that researchers can study the effect of various data filtering techniques. DataComp fixes the computational budget used to train the resulting models, fixing the compute budget of the largest scale to match the cost of training OpenAI's ViT-L/14 CLIP model. These models are then evaluated on a suite of 38 downstream tasks, which includes ImageNet and distribution shifts, VTAB, and retrieval tasks. We use this benchmark as our primary method of evaluating the datasets created by our data filtering networks.

The authors of DataComp also released a baseline dataset, DataComp-1B (DC-1B) that improved upon LAION-5B, by combining CLIP filtering with an ImageNet based clustering approach to improve dataset quality on a variety of benchmarks. However this dataset still relies on the OpenAI CLIP model for CLIP filtering and imposes a costly ImageNet specific clustering step in the pipeline.

Recent work (Xu et al., 2023) has demystified the CLIP dataset construction process and demonstracted high quality dataset construction is possible by simple keyword based sampling and global balancing. While their work does create competitive datasets, the reliance on sampling heuristics from the original CLIP paper (Radford et al., 2021) allows for accurate dataset reproduction, our work focuses on *improving* model performance using dataset construction.

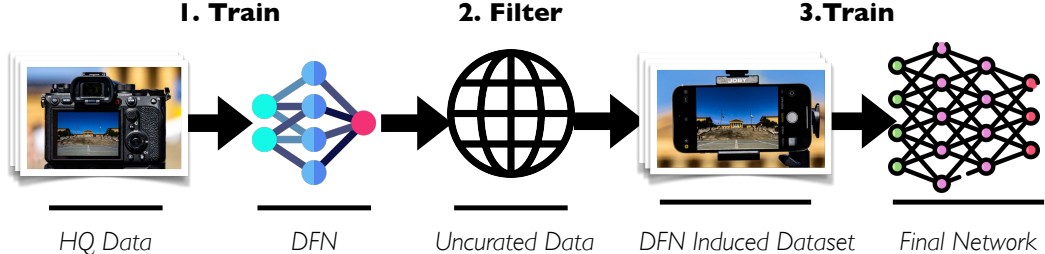

**I. Train**   **2. Filter**   **3.Train**

*HQ Data*   *DFN*   *Uncurated Data*   *DFN Induced Dataset*   *Final Network*

Figure 2: A high level overview of our pipeline for constructing datasets using DFNs

# 3 DATA FILTERING NETWORKS

The core object we study in this work is a data filtering network (DFN). In this section we define DFNs and introduce their evaluation setup.

## 3.1 DEFINITIONS

Since our ultimate goal is to build functions that filter potentially trillions of examples efficiently, we restrict the scope of our study to DFNs that are only applied pointwise to elements of a larger data pool. Thus, processing a data pool with a DFN, defined in pseudocode as follows

```
def apply_dfn(dfn, data_pool):
    return [x for x in data_pool if dfn(x)]
```

lends itself to parallelization and thus efficient application. For a given DFN and pool, we refer to the data pool we train the DFN on as a *filter dataset*. Furthermore, we refer to the dataset constructed by filtering the pool with the DFN the *induced dataset*. We then refer to a model trained (only) on that dataset the *induced model*.

As introduced in Section 2.2 a common choice for a DFN is a CLIP trained image-text model. Thus, a DFN can not only be used to induce a dataset but also be applied to common evaluation problems such as zero-shot ImageNet classification. Inversely, a CLIP model can be both used for general recognition as well as as a DFN. When we use a CLIP model as a DFN, we define its *filtering performance* as the performance of the induced model, as evaluated on standard benchmarks, e.g. ImageNet top-1.

## 3.2 DATA FILTERING NETWORKS EVALUATION SETUP

With these definitions in place, we now address how we evaluate DFNs. In our context, the quality of a DFN is determined by the strength of models it can induce. We build on the evaluation framework proposed by DataComp (Gadre et al., 2023). DataComp provides a multi-scale evaluation framework for datasets by measuring CLIP model zero-shot performance. It provides 4 nested unfiltered image-text pair pools of increasing size. In this work, we use the medium (128M datapoints), large (1.28B datapoints) and xlarge(12.8B datapoints) pools. We also follow the DataComp guidelines of model hyperparameters for each of these pools, which are ViT-B/32 for medium, ViT-B/16 for large and ViT-L/14 for XL. Exact hyperparameters can be found in Table 6. We additionally expand our DFN to a larger pool of 42B images by combining 30B non-DataComp web-scraped images with the DataComp XL pool. We denote the dataset induced using this pool and our DFN as DFN-5B, which we use to train a ViT-H/14 model.

For evaluation we use 38 zero-shot classification and retrieval tasks in the DataComp benchmark. We denote the average performance on these benchmarks simply as "Average" performance, but we also track various subsets: ImageNet performance (IN), ImageNet distribution shift performance (IN shifts), Visual Task Adapation Benchmark (VTAB), Retrieval performance (COCO, Flickr, Wino-GAViL).

Our actual training runs on both Nvidia A100s and TPU v4s. We use OpenClip and AXlearn to train our CLIP models on GPUs and TPUs respectively (Ilharco et al., 2021; Apple, 2023) .

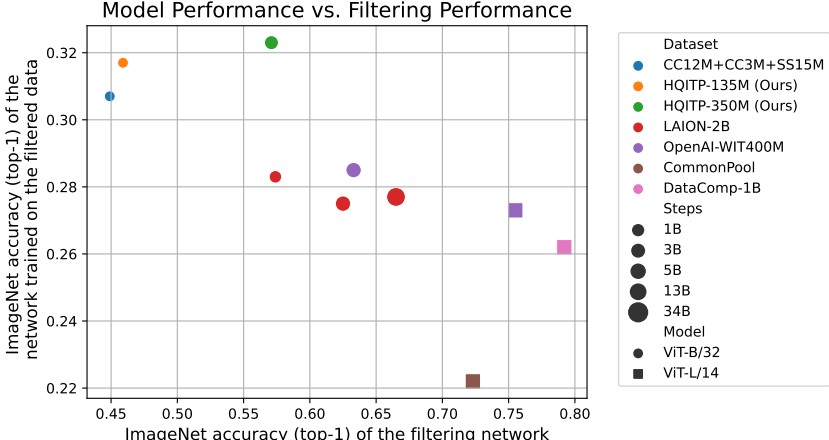

Figure 3: Filtering strength is uncorrelated with image task performance. The models are trained using CLIP, and the number of samples seen and the training data are displayed on the right hand side. Filtering performance is measured by filtering on DataComp medium.

## 3.3 Understanding Data Filtering Networks

As open source CLIP-models improve on standard vision metrics such as ImageNet, the question arises whether we can replace the OpenAI CLIP model used in the dataset construction process with one of these better models. We can even hope of recursively applying this process to continuously train better models that can be used as better filtering models that once again yield even better models. Unfortunately, this does not seem to be true. Figure 3 shows that ImageNet performance of CLIP models is not correlated with filtering performance. To measure filtering performance, we create a dataset by using the CLIP model to apply CLIP filtering on DataComp's medium raw pool, and measure ImageNet performance of models trained on the induced dataset. It is especially striking that a model with 30% less ImageNet performance than OpenAI's CLIP models can be as good when used as a filtering model.

We find that data quality is key to training good filtering models. To demonstrate this, we start with a high-quality pool of 10 million samples from Conceptual 12M (CC12M), and gradually replace it with unfiltered data from Common Crawl until this pool only contains Common Crawl. We train DFNs on these data mixes, and use these DFNs to CLIP filter a separate pool of 128 million Common Crawl samples from DataComp's medium scale. In Figure 4, we measure the ImageNet performance of both the DFNs and the induced models trained on datasets generated by each of the DFNs. While the ImageNet performance of the DFNs degrade steadily as they are trained on larger fractions of unfiltered data, their performance as filtering networks decreases immediately when the high-quality pool is "poisoned" with even a small portion of unfiltered data. Once the filtering training pool is poisoned, the dataset induced by the DFN is only slightly better than unfiltered data.

Table 1: Filtering Performance of various filtering models, after filtering DataComp medium scale (ViT-B/32, 128M samples seen). We present results on ImageNet top-1 as well as "Average" set of tasks (see Sec. 3.2 for details.)

| DFN Type | Filter Dataset | ImageNet | Average |
|---|---|---|---|
| No Filter Baseline | None | 0.176 | 0.258 |
| ResNet-34 Image Binary Filter | ImageNet | 0.242 | 0.292 |
| OpenAI ViT-B/32 Image Binary Filter | ImageNet | 0.266 | 0.295 |
| ResNet-34 Image Binary Filter | CC12M | 0.203 | 0.257 |
| OpenAI ViT-B/32 Image Binary Filter | CC12M | 0.218 | 0.276 |
| M3AE ViT-B/16 | CC12M | 0.237 | 0.297 |
| CLIP ViT-B/32 | CC12M | 0.289 | 0.335 |

Next, we explore using filtering models beyond CLIP models. While DFNs can use any model that can be reduced to a binary function, intuitively it makes sense to use CLIP models. By filtering with

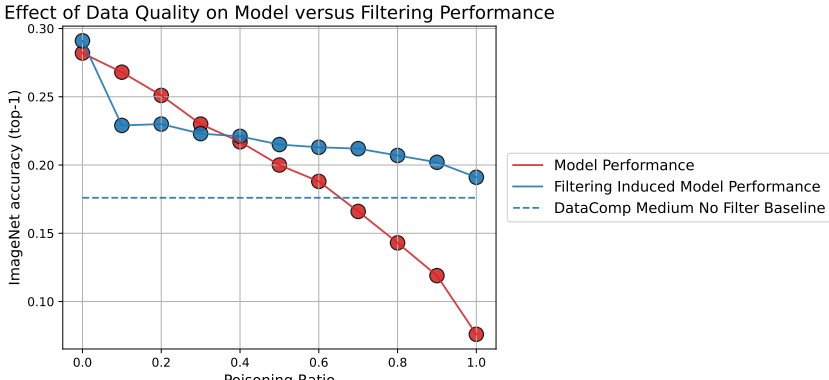

Figure 4: Data quality determines the filtering performance of models. We create these filter training datasets of various quality by having a set pool size of 10 million samples, and interpolating between CC-12M (high quality) and CommonPool (low quality). We then train models induced by the DFN filtering DataComp medium.

a similarity score between the image and text, we encourage keeping samples where the image and text are aligned.

In order to verify this intuition we consider a few other options to produce a DFN. One is to train a binary classifier that can distinguish between ImageNet or CC12M data as positives and Common Crawl as negatives. We consider both ResNet (He et al., 2016) as well as frozen OpenAI CLIP embeddings for this filter. Another option is to use M3AE (Geng et al., 2022) trained on CC12M as a DFN that takes into account both images and text. We can use reconstruction loss as the filtering criterion, as it is a reasonable proxy for how similar samples are to the high-quality data used to train the filtering model.

The filtering performance of all these options, including CLIP models, are summarized in Table 1, where the CLIP model outperform the other backbones. A key difference between the binary classifier and CLIP filters is that the binary filter makes an explicit assumption on what qualifies as a good distribution, while CLIP filters are more flexible. Although the M3AE and CLIP filtering models both are trained on CC12M and examine both modalities, M3AE performs much worse, potentially due to a combination of CLIP encouraging image-text alignment and the difficulty of text reconstruction from just CC12M. We conclude that CLIP models are the most practical and performant models for image-text DFNs.

## 4 CREATING BETTER DATA FILTERING NETWORKS

Equipped with a better understanding of CLIP models as data filtering networks, we aim to create better data filtering networks. DFNs can be trained and modified in the same ways as standard machine learning models. We start by training a CLIP model on a high-quality dataset, and then we can fine-tune the filtering network on subsequent datasets that we want to do especially well on. We use weight ensembling to reduce overfitting on the fine-tuned datasets. Standard machine learning techniques such as augmentation, using a different initialization, and training for more steps with a larger batch size seem to improve the filtering model. We demonstrate the effect of these interventions in Table 2. On the other hand, using a different model size seems to have limited benefits, while model ensembling increases filtering costs without bringing gains. Compared to previous datasets such as DataComp-1B (DC-1B) which involved combining CLIP filtering with clustering-based heuristics, DFNs simplify the data filtering process into a single pipeline while also reducing computational costs.

To create our best DFN, we train a ViT-B/32 CLIP model on High-Quality Image-Text Pairs (HQITP-350M), which is a high-quality dataset of 357 million image-text samples with human-verified captions. This dataset is similar to the HQITP-135M used in Ranasinghe et al. (2023), but expanded to 357M examples. We initialize the weights with OpenAI's checkpoint. We then fine-tune on the combined MS COCO training set, Flickr30k training set, and ImageNet 1k with OpenAI templates as the captions. We use additional augmentation at both training and fine-tuning time.

Table 2: Standard interventions used to improve models can be used on DFNs to induce stronger datasets, leading to better models. DFNs are used to filter and train DataComp large (ViT-B/16, 1.28B samples seen).

| Intervention | | IN | IN Shifts | VTAB | Retrieval | Average |
|---|---|---|---|---|---|---|
| Augmentation | ✗ | 0.620 | 0.493 | 0.534 | 0.515 | 0.536 |
| | ✓ | 0.626 | 0.501 | 0.534 | 0.516 | 0.542 |
| Samples Seen / Batch Size | 2.56B / 4096 | 0.626 | 0.506 | 0.536 | 0.511 | 0.545 |
| | 5.12B / 8192 | 0.624 | 0.508 | 0.551 | 0.517 | 0.550 |
| Fine-tune | ✗ | 0.624 | 0.508 | 0.551 | 0.517 | 0.550 |
| | ✓ | 0.678 | 0.540 | 0.555 | 0.534 | 0.560 |
| OAI-Init | ✗ | 0.674 | 0.535 | 0.533 | 0.529 | 0.548 |
| | ✓ | 0.678 | 0.540 | 0.555 | 0.534 | 0.560 |

Table 3: Training on DFN-2B produces state-of-the-art CLIP models. Here we evaluate on the DataComp benchmark, comparing against LAION-2B, DC-1B, MetaCLIP and OpenAI WIT-400M. Additional comparisons can be found on the DataComp leaderboard.

| Dataset | DataComp Scale | IN | IN Shifts | VTAB | Retrieval | Average |
|---|---|---|---|---|---|---|
| DC-1B | medium | 0.297 | 0.239 | 0.346 | 0.231 | 0.328 |
| DFN-2B | medium | 0.371 | 0.298 | 0.388 | 0.288 | 0.373 |
| DC-1B | large | 0.631 | 0.508 | 0.546 | 0.498 | 0.537 |
| DFN-2B | large | 0.678 | 0.540 | 0.555 | 0.534 | 0.560 |
| LAION-2B | xlarge | 0.731 | 0.603 | 0.586 | 0.589 | 0.601 |
| OpenAI WIT-400M | xlarge | 0.755 | 0.649 | 0.586 | 0.543 | 0.617 |
| DC-1B | xlarge | 0.792 | 0.679 | 0.652 | 0.608 | 0.663 |
| DFN-2B | xlarge | 0.814 | 0.688 | 0.656 | 0.649 | 0.669 |
| LAION-2B | ViT-G/14-224px | 0.801 | 0.691 | 0.646 | 0.635 | 0.667 |
| DC-1B (CLIPA-v2) | ViT-G/14-224px | 0.831 | **0.740** | 0.645 | 0.631 | 0.684 |
| MetaCLIP | ViT-H/14-336px | 0.805 | 0.700 | 0.640 | 0.652 | 0.667 |
| WebLI | ViT-SO/400M-384px | 0.831 | 0.734 | 0.648 | **0.698** | 0.692 |
| DFN-2B | ViT-L/14-224px | 0.822 | 0.679 | 0.664 | 0.666 | 0.678 |
| DFN-5B | ViT-H/14-224px | 0.834 | 0.713 | 0.675 | 0.684 | 0.698 |
| DFN-5B | ViT-H/14-378px | **0.844** | 0.738 | **0.685** | 0.695 | **0.710** |

Additional training details can be found in Appendix B. We create our dataset DFN-2B by applying this DFN on DataComp's full 12.8 billion sample CommonPool, with a threshold equivalent to taking the top 15% of samples.

Our DFN induces datasets that achieve state-of-the-art results on medium, large, and xlarge scales in DataComp. In particular at xlarge, we train a ViT-L/14 on DFN-2B for 12.8B samples seen to achieve 81.4% zero-shot accuracy on ImageNet, and a 0.669 average over 38 DataComp evaluation datasets. As shown in Table 3, in terms of ImageNet zero-shot improvement, this is a 2.2% improvement over DC-1B, a 5.9% improvement over OpenAI WIT-400M, and a 8.3% improvement over LAION-2B. These improvements are beyond ImageNet, as we can see similar trends across the DataComp evaluation suite in distribution shifts, retrieval, VTAB, and average performance. We also train a ViT-L/14 on DFN-2B for 39B samples seen. Lastly, we train DFN-5B on a ViT-H/14 for 39B samples seen at 224 × 224 resolution, and 5B samples at 378 ×378 resolution – achieving 84.4% zero-shot transfer accuracy on ImageNet, and 0.710 average on the DataComp evaluation suite. We find that models trained our DFN produced datasets outperform all other models on the evaluation suite regardless of pre-training dataset: MetaClip, WebLI or DataComp-1B (Xu et al., 2023; Zhai et al., 2022a; Gadre et al., 2023), archifectural improvements such as shape-optimized ViTs (Alabdulmohsin et al., 2023), a more performant sigmoid loss (Zhai et al., 2023), or pre-training performance optimizations such as those in Li et al. (2023b).

Table 4: High-quality data is best used to train the filtering model rather than the end model. Training DFNs with HQITP-350M induces a dataset that outperforms the dataset induced by a worse DFN combined with HQITP-350M. Additional experiments with a ViT-B/16 are in Appendix Table 10.

| Dataset | Model | IN | IN Shifts | VTAB | Retrieval | Average |
|---|---|---|---|---|---|---|
| OAI ViT-B/32 Induced Dataset + HQITP-350M | ViT-L/14 | 0.774 | 0.654 | 0.643 | 0.616 | 0.654 |
| DFN without FT Induced Dataset | ViT-L/14 | 0.787 | 0.670 | 0.654 | 0.631 | 0.666 |
| DFN-2B | ViT-L/14 | 0.814 | 0.688 | 0.656 | 0.649 | 0.669 |
| DFN-2B + HQITP-350M | ViT-L/14 | 0.813 | 0.691 | 0.662 | 0.656 | 0.670 |

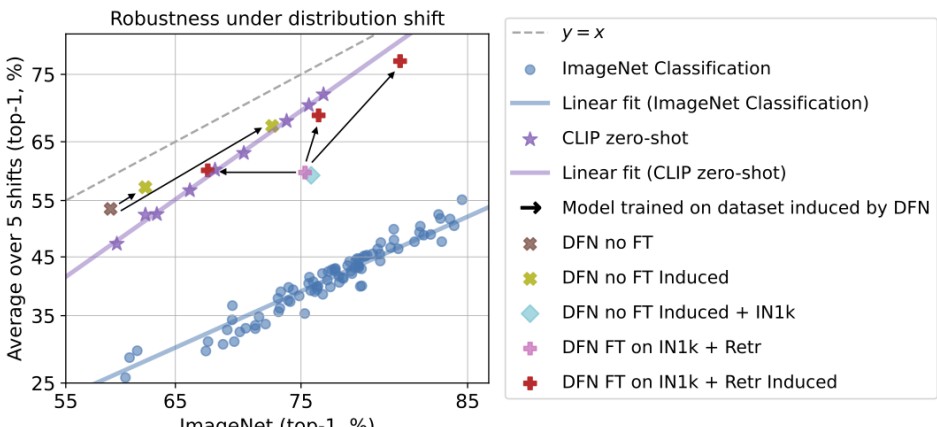

Figure 5: Datasets induced by DFNs can be robust to distribution shift. DFNs can be fine-tuned to maintain robustness of induced datasets, unlike directly training on ImageNet (IN). DFNs are not performing distillation because induced datasets lead to higher performing models than the original DFN. Distribution shifts used are IN-V2, ObjectNet, IN-Sketch, IN-R, and IN-A (Recht et al., 2019; Barbu et al., 2019; Wang et al., 2019; Hendrycks et al., 2021a;b).

Creating better datasets not only improves model performance, but also improves model efficiency. Performance that was once only achievable by larger models can be matched with a smaller model trained on a better dataset. Our ViT-L/14 trained on DFN-2B surpasses a ViT-G/14 trained on LAION-2B for 34B samples seen by 1.5% zero-shot accuracy on ImageNet, and by 0.002 average performance, despite using 16x less computational cost [2]. Similarly, we can train a ViT-B/16 on DFN-2B for 12.8B samples seen to achieve competitive performance with OpenAI's ViT-L/14, representing a 4x computational cost reduction.

The key to training good DFNs is using high-quality data for training the filtering network. Collecting verified high-quality data is expensive, as it often requires human annotations, and is thus difficult to scale to large quantities. But given a sizable high-quality dataset, we can explore if there are benefits to directly training on it instead of using it to train a DFN. In Table 4, we compare models trained on datasets induced by our DFNs with a model trained on HQITP-350M combined with the dataset induced by CLIP filtering CommonPool with OpenAI's ViT-B/32. Models trained on DFN induced datasets outperform the baseline on all major categories within the DataComp evaluation suite. Furthermore, training on the combination of HQITP-350M and DFN-2B seems to have little improvement when compared to just training on DFN-2B. By training a DFN instead of directly training on high-quality data, we demonstrate a successful recipe for leveraging high-quality data for creating large-scale high-quality datasets.

We can also explore the differences between fine-tuning a DFN and directly training on the fine-tuning dataset. In Figure 5 and Table 7, we compare models trained on a dataset induced by a baseline DFN, a dataset induced by the baseline DFN fine-tuned on ImageNet and a dataset induced by the baseline DFN without fine-tuning on ImageNet combined with ImageNet. While the model

---

[2]calculation does not take into account patch dropout used to train ViT-G/14 on LAION-2B

that directly trains on ImageNet has much higher performance on ImageNet and ImageNet-V2, it does not improve upon the baseline for the ObjectNet, ImageNet-Sketch, and ImageNet-R. On the other hand, the DFN fine-tuned on ImageNet induces a dataset that improves over the baseline on ImageNet and all of its distribution shifts. Fine-tuning on DFNs acts as a regularizer to induce datasets similar to the fine-tuning dataset, while maintaining strong robustness properties that come with drawing from a more distributionally diverse candidate pool.

## 4.1 BETTER DATASETS BEYOND VISION TASKS: VQA

Just like how machine learning models would ideally generalize across many tasks, we would also like our datasets to generalize across diverse tasks. We show that our datasets not only lead to better models when evaluated on vision tasks, but also lead to better visual question answering (VQA) models. We train a BLIP2 model (Li et al., 2023a) which takes as input a CLIP visual encoder and is trained for zero-shot VQA on COCO and Visual Genome, to measure zero-shot VQA performance on VQVA2, GQA, and OKVQA (Goyal et al., 2017; Hudson & Manning, 2019; Marino et al., 2019). We compare the performance on BLIP2 between the standard OpenAI ViT-L visual encoder and the ViT-L trained on DFN-2B. The DFN-2B model consistently outperforms the OpenAI ViT-L encoder and is competitive with a much larger EVA ViT-g model trained on LAION-2B[3]

Table 5: Performance of BLIP-2 variants with different visual encoder training datasets. The DFN-2B trained ViT-L provides consistent improvements across multiple zero-shot VQA tasks.

| Visual Encoder Training Dataset | Architecture | VQAv2 Acc. (%) | GQA Acc. (%) | OKVQ Acc. (%) |
|---|---|---|---|---|
| OAI-WIT-400M | ViT-L | 45.5 | 30.0 | 19.1 |
| DFN-2B | ViT-L | 48.3 | 31.3 | 21.9 |
| LAION-2B | ViT-g | 48.7 | 31.1 | 24.5 |

## 4.2 PUBLICLY REPRODUCIBLE DFNS

Scientific research benefits from results that can be reproduced by anyone from scratch. Though OpenAI's internal dataset and HQITP-350M are not publicly accessible, we demonstrate that a competitive DFN can be trained on public data sources. We train a ViT-B/32 on Conceptual Caption12M, Conceptual Captions 3M, and Shutterstock 15M (Changpinyo et al., 2021; Sharma et al., 2018; Nguyen et al., 2023). As shown in Appendix Table 11 , this DFN surpasses OpenAI's ViT-B/32 in terms of filtering performance at DataComp's medium, large, and xlarge scales. Additionally, this DFN can be modified as described in the previous section to further improve filtering performance.

## 5 DISCUSSION

The simplicity of the data filtering network pipeline makes it a flexible tool to integrate into existing workflows. As DFNs operates on individual samples, this approach scales linearly with candidate pool size, enabling the creation of datasets orders of magnitude larger than those that we introduce in this work. Additionally, the DFNs we train in this work are relatively small neural networks which allows for filtering to be directly integrated into training procedures of much larger networks for minimal marginal cost. DFNs can then filter batches of raw data that are then trained on, reducing the need for complex data pre-processing procedures.

As useful as DFNs are in building performant models in this work, there are still many unanswered questions to address in future work. We still do not know exactly how to optimize directly for dataset quality, and thus opt for weak proxies such as alignment. It is not even clear what that proxy would be for other domains where DFNs could be applied such as speech, text or video data. We hope that these open questions and the bridge DFNs build between modeling work and dataset work can lead to fruitful new avenues of research.

---

[3]EVA's ViT-g has an additional pre-training procedure trained on ImageNet-21k, COCO, Objects365 and Conceptual Captions 12M

## ACKNOWLEDGEMENTS

We would like to thank Bowen Zhang, Ruoming Pang, Brandon McKinzie, Mitchell Wortsman, Gabriel Ilharco, Ross Wightman, Achal Dave, Josh Susskind, Alaaeldin Ali, Fartash Faghri, Preetum Nakkiran, and Chen Huang for helpful feedback at various stages of the project.

Bowen and Ruoming were invaluable for helping us set up AxLearn and answering countless questions when we ran into various errors. Brandon pointed us in the direction of the HQITP datasets that were crucial for our best results, and also helped us with AxLearn issues. Mitchell's OpenClip experience helped us set hyper-parameters for our largest scale runs. Gabriel helped us debug a webdataset related dataloader bug. Ross caught a bug in our final high resolution model that led to a modest performance improvement. Achal provided hyper-parameters and instructions for the BLIP2-based VQA experiments. Alaaeldin, Chen, Fartash, Josh, and Preetum provided helpful comments on the manuscript.

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

# A    TRAINING HYPERPARAMETERS

Table 6: We follow the hyperparameter settings of the DataComp paper for the medium, large and xlarge pool

| Dataset | Model | Pool size and # seen samples | Batch Size | Max LR | Weight Decay | Warmup | Beta2 |
|---------|-------|------------------------------|------------|--------|--------------|--------|-------|
| DataComp-medium | ViT-B/32 | 128M | 4096 | 5e-4 | 0.2 | 500 | - |
| DataComp-large | ViT-B/16 | 1.28B | 8192 | 5e-4 | 0.2 | 500 | - |
| DataComp-xlarge | ViT-L/14 | 12.8B | 90112 | 1e-3 | 0.2 | 10000 | 0.95 |
| DFN-5B-pool | ViT-H/14 | 39B | 79872 | 2e-3 | 0.1 | 10000 | 0.95 |

# B    DFN HYPERPARAMETERS

DFNs trained for ablations use DataComp large scale hyperparameters with a ViT-B/32 instead of a ViT-B/16. Final DFNs that induce DC-2B train for 5.12B samples, 16,384 batch size, and 2,000 steps of warmup.

# C    ROBUSTNESS OF USING IMAGENET AT FILTERING VS. TRAINING TIME

Table 7: Fine-tuning a DFN on ImageNet induces datasets with nice robustness properties that are lost when directly training on ImageNet. Ran at DataComp large scale (ViT-B/16, 1.28B samples).

| Dataset | IN | IN-V2 | ObjectNet | IN-Sketch | IN-R | IN-A | VTAB |
|---------|-----|-------|-----------|-----------|------|------|------|
| Baseline DFN | 0.624 | 0.547 | 0.511 | 0.510 | 0.724 | 0.257 | 0.551 |
| Baseline DFN FT on ImageNet | 0.678 | 0.594 | 0.536 | 0.536 | 0.743 | 0.284 | 0.555 |
| Baseline DFN + IN | 0.757 | 0.652 | 0.509 | 0.512 | 0.703 | 0.272 | 0.543 |

# D    FULL EXPERIMENTAL EVALUATION & MODEL RELEASE

Below we provide links to checkpoints and detailed evaluation results of models in Table 3 on each of the 38 DataComp evaluation datasets

| Model Link | ImageNet | Average |
|------------|----------|---------|
| DFN5B-CLIP-ViT-H-14-378 | 0.844 | 0.710 |
| DFN5B-CLIP-ViT-H-14 | 0.834 | 0.698 |
| DFN2B-CLIP-ViT-L-14 | 0.814 | 0.669 |
| DFN2B-CLIP-ViT-B-16 | 0.762 | 0.609 |

Table 8: Links to checkpoints and detailed evaluation results

# E  FIGURES MEASURING AVERAGE PERFORMANCE INSTEAD OF IMAGENET

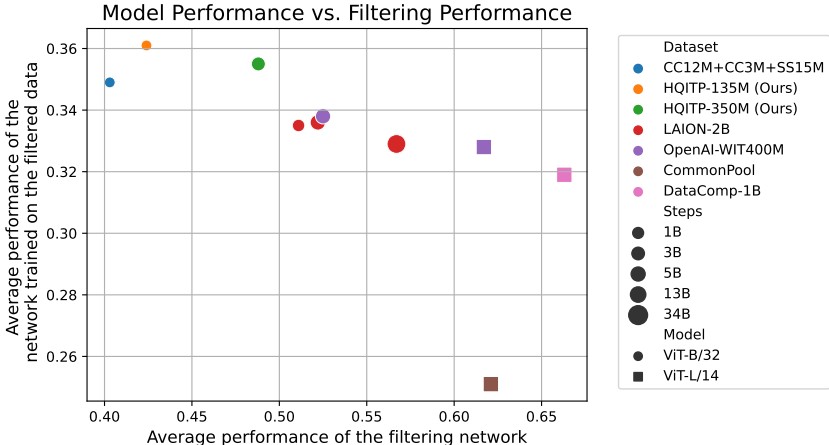

Figure 6: Average accuracy version of Figure 3.

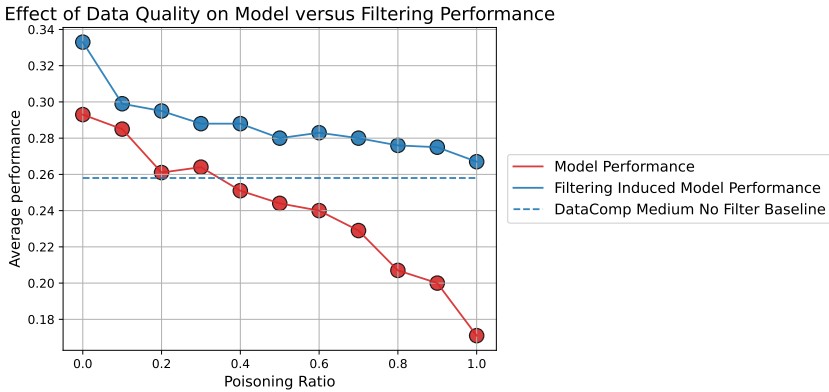

Figure 7: Average accuracy version of Figure 4.

# F    Log-Scale plot of Figure 1

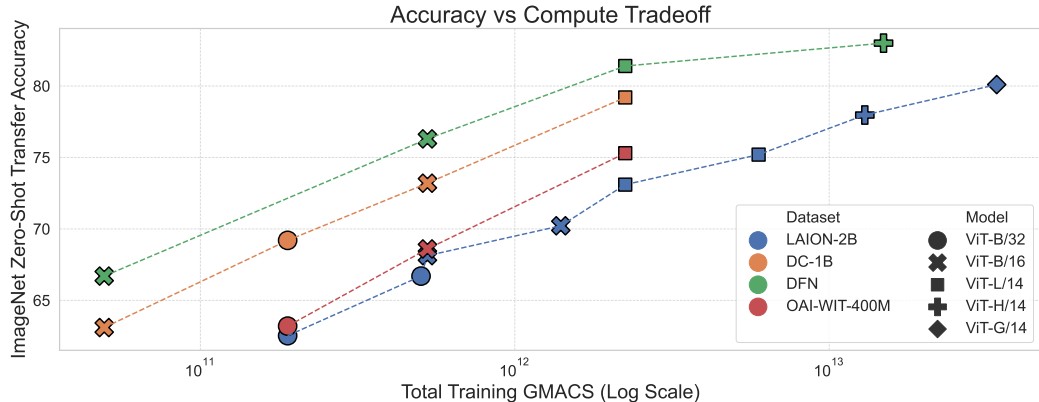

Figure 8: Compute scaling behavior of training CLIP models on various datasets (log scale)

# G    Additional Tables

Table 9: We can produce high-quality DFNs completely from scratch. Specifically, we do not use any OpenAI CLIP models for results in this table. We also use HQITP-135M for DFN training, a subset of HQITP-350M that we use in the rest of the paper

| Dataset | DataComp Scale | IN | IN Shifts | VTAB | Retrieval | Average |
|---|---|---|---|---|---|---|
| Induced by DFN HQITP-135M with FT on IN1k, no OAI-Init, no Aug., samples 1B, BS 4096 | xlarge | 0.805 | 0.665 | 0.641 | 0.639 | 0.663 |

Table 10: High-quality data is best used to train the filtering model rather than the end model. Training DFNs with HQITP-350M induces a dataset that outperforms the dataset induced by a worse DFN combined with HQITP-350M.

| Dataset | Model | IN | IN Shifts | VTAB | Retrieval | Average |
|---|---|---|---|---|---|---|
| OAI ViT-B/32 Induced Dataset + HQITP-350M | ViT-B/16 | 0.706 | 0.572 | 0.582 | 0.575 | 0.596 |
| DFN without FT Induced Dataset | ViT-B/16 | 0.729 | 0.599 | 0.604 | 0.597 | 0.612 |
| DFN-2B | ViT-B/16 | 0.762 | 0.623 | 0.598 | 0.611 | 0.609 |
| OAI ViT-B/32 Induced Dataset + HQITP-350M | ViT-L/14 | 0.774 | 0.654 | 0.643 | 0.616 | 0.654 |
| DFN without FT Induced Dataset | ViT-L/14 | 0.787 | 0.670 | 0.654 | 0.631 | 0.666 |
| DFN-2B | ViT-L/14 | 0.814 | 0.688 | 0.656 | 0.649 | 0.669 |
| DFN-2B + HQITP-350M | ViT-L/14 | 0.813 | 0.691 | 0.662 | 0.656 | 0.670 |

## H CLIP FILTERING PSEUDOCODE

We show pseudocode of the basic CLIP filtering operation below.

```python
def clip_filter(image, text, threshold=0.3):
    # compute image and text representations
    image_features = clip.encode_image(image_input)
    text_features = clip.encode_text(text_input)
    # compute alignment
    dot_product = image_features.T @ text_features
    norm_a = image_features.norm()
    norm_b = text_features.norm()
    similarity = dot_product / (norm_a * norm_b)
    # filter by alignment
    return similarity > threshold
```

## I PUBLICLY REPRODUCIBLE DFNS

Table 11: DFNs are trained with a ViT-B/32, then used to filter DataComp pools. Conceptual 12M, Conceptual Captions 3M, and Shutterstock 15M are publicly available datasets, demonstrating that large-scale high-quality datasets can be constructed with only publicly available resources.

| DFN Training Data | DataComp Scale | IN | IN Shifts | VTAB | Retrieval | Average |
|---|---|---|---|---|---|---|
| CC12M + CC3M + SS15M | medium | 0.307 | 0.253 | 0.359 | 0.274 | 0.349 |
| OpenAI WIT-400M | medium | 0.285 | 0.240 | 0.355 | 0.253 | 0.338 |
| CC12M + CC3M + SS15M | large | 0.591 | 0.481 | 0.522 | 0.503 | 0.532 |
| OpenAI WIT-400M | large | 0.578 | 0.466 | 0.525 | 0.475 | 0.527 |
| CC12M + CC3M + SS15M | xlarge | 0.770 | 0.656 | 0.663 | 0.624 | 0.658 |
| OpenAI WIT-400M | xlarge | 0.764 | 0.640 | 0.628 | 0.599 | 0.638 |

