# OpenReview forum: "Data Filtering Networks"
_ICLR.cc/2024/Conference — ICLR 2024 poster_

### Official Review · Reviewer_ZvvG · 2023-10-22

**Soundness:** 3 good
**Presentation:** 3 good
**Contribution:** 3 good
**Rating:** 6
**Confidence:** 5

**Summary:**

The paper proposes a contrastive loss trained model for data filtering

**Strengths:**

1. The finding is interesting that a model's data filtering ability is not correlated with its image task performance. Actually, which is also discussed by [1].
2. The finding is interesting and useful that data quality determines the trained model's filtering performance.
3. The proposed data filtering network achieves impressive results on data filtering.

[1] Yu, Haichao, et al. "The Devil is in the Details: A Deep Dive into the Rabbit Hole of Data Filtering." arXiv preprint arXiv:2309.15954 (2023).

**Weaknesses:**

1. The HQITP-350M dataset is the key to the success of the DFN, so it is disappointing that it cannot be publicly accessible by the research community.

**Questions:**

1. How many runs are conducted to get the quantitative experimental results in the paper (e.g., Table 1-6)? Are the standard deviations sufficiently smaller than the differences between different settings?

---

> ### Author Response · Authors · 2023-11-17
> **Response to reviewer ZvvG**
>
> Thank you for your review. We are happy that you find our work interesting and the results impressive. We hope to address your remaining concerns.
>
> To address your questions and concerns about HQITP we provide a copy of our general response from above.
> We provide some clarity about the HQITP-350M dataset and what we omitted from the paper.
>
> * While we cannot release HQITP-350M for legal purposes, we have shown it is possible to build a DFN better than the existing OAI-CLIP models using only publicly available data. While we only did this up to the “large” DataComp scale in the original submission, we extend this to the XL scale below and compare it to the previous state of the art filtering network (OAI-CLIP-B/32). Our DFN with none of the additional modifications (fine-tuning/open-ai-init/augmentation) from section 4 in the paper outperforms the OAI DFN.
> * Because HQITP-350M dataset would be similar in nature to licensed stock image datasets, so we don’t have access to the exact labeling decision behind each image caption. We also note we will release the dataset induced by our best performing DFN which is sufficient for the community to train strong models.
> * In table 2 (above), We provide an additional ablation below where we control the number of examples from HQITP used to train the DFN and measure its performance on the datacomp benchmark (at the medium scale). We show that one can achieve most of the gains with a fraction of the of the full dataset. We note this experiment was done with an earlier version of HQITP that only contained 135M examples.
>
>
> Each result is a single run, as these experiments are quite expensive at the highest scales. However, we can provide error bars for the results on the validation tasks - below in table 5 are 95% Clopper-Pearson confidence intervals for the results of our best model.

---

> > ### Author Response · Authors · 2023-11-17
> > **Table 5**
> >
> > | eval dataset                             | accuracy            |
> > |:------------------------------------|:--------------------|
> > | cifar10                             | 0.988 (0.986-0.990) |
> > | vtab/cifar100                       | 0.904 (0.898-0.910) |
> > | vtab/clevr_count_all                | 0.362 (0.355-0.370) |
> > | vtab/clevr_closest_object_distance  | 0.206 (0.200-0.213) |
> > | country211                          | 0.377 (0.370-0.383) |
> > | vtab/dtd                            | 0.714 (0.693-0.734) |
> > | vtab/eurosat                        | 0.608 (0.595-0.621) |
> > | food101                             | 0.963 (0.961-0.965) |
> > | gtsrb                               | 0.679 (0.671-0.687) |
> > | imagenet1k                          | 0.842 (0.839-0.845) |
> > | imagenet_sketch                     | 0.733 (0.730-0.737) |
> > | imagenetv2                          | 0.784 (0.775-0.792) |
> > | imagenet-a                          | 0.799 (0.790-0.808) |
> > | imagenet-o                          | 0.379 (0.357-0.400) |
> > | imagenet-r                          | 0.938 (0.935-0.940) |
> > | vtab/kitti_closest_vehicle_distance | 0.383 (0.347-0.419) |
> > | mnist                               | 0.837 (0.830-0.844) |
> > | objectnet                           | 0.797 (0.791-0.803) |
> > | voc2007                             | 0.826 (0.820-0.832) |
> > | vtab/pcam                           | 0.696 (0.691-0.701) |
> > | renderedsst2                        | 0.567 (0.544-0.590) |
> > | vtab/resisc45                       | 0.755 (0.744-0.766) |
> > | cars                                | 0.960 (0.955-0.964) |
> > | stl10                               | 0.991 (0.989-0.993) |
> > | sun397                              | 0.773 (0.770-0.775) |
> > | vtab/svhn                           | 0.671 (0.666-0.677) |
> > | wilds/camelyon17                    | 0.712 (0.708-0.715) |

---

> > > ### Author Response · Authors · 2023-11-22
> > > **Response**
> > >
> > > As the reviewer-author discussion period ends tomorrow, we kindly ask if the reviewer has had time to read through our rebuttal. We addressed both points the reviewer raised regarding HQITP-350M and the lack of error bars, and it would be helpful to hear the reviewer’s response.

---

> > > ### Comment · Reviewer_ZvvG · 2023-11-22
> > >
> > > Thanks for the detailed comments and additional results.

---

### Official Review · Reviewer_2tx6 · 2023-10-28

**Soundness:** 3 good
**Presentation:** 3 good
**Contribution:** 1 poor
**Rating:** 5
**Confidence:** 5

**Summary:**

This paper's focus is on dataset filtering for contrastive language-image pre-training (CLIP). The discussion on data-filtering networks (DFN) is motivated by the use of pretrained CLIP models (referred to as the DFN) to filter datasets by discarding image-caption pairs whose embeddings (as produced by the DFN) are not similar to each other. The paper's main observation is that accuracy of the DFN on a given task doesn't predict how good a subset it selects (even for that task). The paper then uses this to motivate the creation of new DFNs that are trained on high quality image-caption data and then fine-tuned on "important" datasets. The paper confirms the effectiveness of the approach by demonstrating competitive performance on ImageNet Zero-Shot, ImageNet Distribution Shift, VTAB, Retrieval etc when filtering DataComp's medium, large and xlarge data.

**Strengths:**

1. The problem of better understanding and improving DFNs is extremely interesting.

2. The observation that a DFN with good accuracy on a given task doesn't necessarily induce a good dataset on that task is very interesting.

3. Several large-scale experiments are conducted to explore the effectiveness of the approach.

**Weaknesses:**

It seems like the improvement in performance over other DFNs is primary in ImageNet (and dist. shift) Zero-Shot as well as Retrieval. It seems like without "finetuning" on MS COCO training set, Flickr30k training set, and ImageNet 1K training set with ImageNet templates as captions, these performance improvements nearly disappear. While the main paper does ablate over fine-tuning, it doesn't highlight just how significant fine-tuning on these particular datasets is (primarily because these are the datasets we wish to evaluate on).

Personally, I don't find the conclusion that training a DFN on high-quality image-caption datasets allows the induced subsets to be higher quality very surprising or interesting. I do think a fine-grained study into specific properties of the pretraining data (e.g. similarity with downstream tasks' data, diversity of data, data imbalance) can make this work interesting and help further our understanding of DFNs. In it's current form, apart from the observation on ImageNet accuracy of DFN and that of the model trained on the induced dataset, I don't see what the contribution of this work is towards understanding DFNs.

While I recognize, the improvement in accuracy on ImageNet, Retrieval etc., the importance of fine-tuning the DFN on the evaluation datasets leads to me believe that this approach may not generalize to other downstream tasks (as evidenced by the < 1% improvement on VTAB).

**Questions:**

1) Is there any explanation for why a DFN with higher ImageNet accuracy is not always better at selecting data for ImageNet?

2) Could the authors go into greater depth about the impacts of fine-tuning i.e. is the key component of training a good DFN fine-tuning on the evaluation datasets. If so, how do the authors see these models as generalizing on unseen tasks?

3) Is there a trade-off in accuracy on some datasets? For one of the models, could we see the per dataset accuracy for VTAB using different DFNs and see if there are trade-offs in which datasets improve and which don't?

---

> ### Author Response · Authors · 2023-11-17
> **Response to reviewer 2tx6**
>
> Thank you for your detailed review. We hope to address your concerns and are happy to answer any additional questions.
>
>
> While some may not find these results surprising, we believe this is the first work that trains a CLIP model specifically for the purpose of filtering data for large-scale dataset creation, whereas all prior work relies on using an OpenAI CLIP model to filter the data.
>
> In the paper we provided analysis for understanding DFNs beyond showing that better ImageNet CLIP models do not result in better filtering models. We demonstrated that data quality is more important for training filtering models than for training standard models. Adding model filtered data from a raw pool is a noisy process that can still include bad training points, and these points negatively affect a model used for filtering much more than it affects standard models. For additional details, see Figure 4 and Section 3.3 in the paper.
>
>
> ## Overfitting to ImageNet/COCO/FLICKR
> We understand that the reviewer is concerned about potential overfitting to ImageNet because our filtering network trains on ImageNet + Coco + Flickr (while our data pool is de-duplicated against these datasets). To address this we first provide a full results table to provide an in depth analysis of results and provide ablations at a higher compute scale that show even less signs of overfitting to ImageNet/COCO/Flickr. We also add that while fine-tuning does help performance on a set of key benchmarks, our non-finetuned DFN consistently outperforms the best and most widely used DFN today (the OAI-CLIP-ViT-B//32).
> We apologize for only providing averages of our evaluation which could cloud some of the takeaways. We provide a full results table (Table 3 and 4 above) for our best performing and comparable external models on the full evaluation suite. We will update our final manuscript to have links to these full results tables. We would like to highlight three points from this full results analysis
> 1. Performance on several of the VTAB tasks (Cifar-10, Caltech-101, STL-10)  are close to saturated so making large improvements across all the tasks can be difficult, and even a 0.5% average improvement can be meaningful. We do agree that performance on a few high quality benchmarks may be more indicative of performance improvements.
> 2. Our benchmark contains 2 additional high quality test sets specifically for highlighting biases in traditional imagenet style models DollarStreet and GeODE, this is accounted for in the average calculation but not VTAB calculation.
> 3. We note the retrieval benchmark average is an average of 3 tasks (Flickr, Coco and WinoGAVIL). WinoGAVIL was a retrieval benchmark released in 2022 containing images from google images and de-duplicated from our training set.
>
> With the in-depth results table in mind, we provide 3 additional models that provide evidence that overfitting to ImageNet/CoCO/Flickr is not a significant concern.
> 1. First we scaled up our ViT-H/14 method further with 1 epoch of high resolution fine-tuning and compare our results across all categories to a state of the art CLIP model that performs inference at a higher resolution: SiGLIP (particularly the ViT-SO400M-14-SigLIP variant). We show that not only does our approach out-perform SiGLIP on ImageNet (84.4 vs 83.1) and average across the 38 benchmarks (71.0 vs 69.4), our performance gap on VTAB is greater than that on ImageNet or Retrieval (68.5 vs 64.6). We show strong performance on DollarStreet and GeODE, slightly under-performing SiGLIP on the former, and over-performing on the latter. We additionally note that this is all with the SigLIP model using a more efficient architecture [2] and being trained using the same compute budget.
> 2. We agree our ViT-B/16 results seem to indicate we are “weakening” the model somehow by training the DFN on ImageNet, we show that at larger scale this phenomena reverses. We provide results of a ViT-L at XL datacomp-scale induced by the DFN without the fine-tuning on ImageNet/COCO/FLICKR. We see that the *average* performance of the model is higher for the finetuned-DFN model compared to the “base” one (that does not finetune on IN). In addition we see improvements in both WinoGAVIL (+3pp) , DollarStreet(+2pp) and a slight decrease on GeoDE (-0.7pp).
> 3. Finally we take our XL ViT-L model and continue training it for 3x the number of iterations (39B examples seen) and we see that while we get a slight performance win on ImageNet (+0.7pp), we get a larger gain on other aspects of our benchmark such as  +1.2pp on average, +0.9pp on VTAB, +1.5pp on Geode and a slight performance degradation on DollarStreet (-0.5pp).

---

> > ### Comment · Reviewer_2tx6 · 2023-11-19
> >
> > Thank you for responding to my comments with specific experiments.
> >
> > As mentioned in my original review, I find the problem of studying DFNs an extremely valuable direction of research and am glad to see the authors for shedding light on some empirical benefits of this. From my understanding of the review, the authors have clarified that the evaluation data is de-duplicated from the training data of the DFN. This was a major concern for me originally and I'm glad to see that this is not the case.
> >
> > I find that the conclusions of the paper i.e. "better data -> better DFN" to be an oversimplification of the underlying phenomena. From seeing the trends on (fine-tuning with ImageNet etc. and not) and the effects on the performance of different datasets, it seemed to be that if the DFN is trained / fine-tuned on data that is similar to the evaluation data, it would select better data for that task. While these numbers seem to change with the larger-scale experiments run by the authors, I'm not confident that "better data -> better DFN" accurately describes what is happening in these experiments.
> >
> > With that, I stand by my original review and I encourage the authors to dive a little deeper into the relations between what kind of data the DFN is trained on and how that affects performance on specific downstream tasks.

---

> ### Author Response · Authors · 2023-11-19
> **Response to Reviewer 2tx6’s response**
>
> We thank the reviewer for their quick response. It would be helpful for us to understand the reviewer’s concerns better. For instance, the reviewer writes
>
> > I'm not confident that "better data -> better DFN" accurately describes what is happening in these experiments.
>
> Figure 4 shows how adding even a small amount of noisy data into the DFN training set leads to worse induced models - hence “better data -> better DFN”. What aspect of this experiment does the reviewer not agree with?
>
> Finally, we kindly ask that the reviewer re-consider their score as there now seems to be agreement on multiple positive aspects of our submission:
> * The reviewer calls DFNs “an extremely valuable direction of research”
> * The reviewer agrees that we have addressed the concern regarding overfitting
> * Our experimental results are the best published CLIP-style models in the literature.
>
> What remains is that the reviewer encourages us “to dive a little deeper into the relations between what kind of data the DFN is trained on and how that affects performance on specific downstream tasks.” We agree that this is an interesting direction, and plan to investigate it in future work. But in one paper it is unfortunately not possible to follow all potential follow-up questions to their end. So taken together, we wonder if the overall assessment / rating is still “3: reject, not good enough”

---

> > ### Comment · Reviewer_2tx6 · 2023-11-20
> > **Increased score from 3 -> 5**
> >
> > I appreciate the authors' points and have increased the score from 3 -> 5 in light of the further experiments on the fine-tuning on ImageNet etc.
> >
> > The reason I still do not recommend this paper for acceptance is that I don't feel in the current form the findings are significant / surprising / nuanced enough to be a completed paper (despite the problem statement being quite interesting). I understand this is a potentially subjective opinion, thus I've increased the score to allow the AC to make a better decision.

---

> > > ### Author Response · Authors · 2023-11-22
> > > **Thank you!**
> > >
> > > We thank the reviewer for their quick response and raising their score! We will look closer at the relationship between DFN training data and downstream task performance. Any concrete suggestions for experiments would be welcome!

---

### Official Review · Reviewer_Q5jf · 2023-10-31

**Soundness:** 4 excellent
**Presentation:** 3 good
**Contribution:** 4 excellent
**Rating:** 8
**Confidence:** 4

**Summary:**

The paper studies the problem of how to filter large-scale web-scrawled data into the training data for training CLIP models. The paper proposes a data filtering network (DFN) for this task.
* Training DFN: DFN is a small CLIP model pretrained on a small but “high-quality” dataset based on the authors’ insight: (1) the performance of DFN itself is not correlated to the induced model trained on DFN’s filtered data (Figure 3) and (2) Induced model’s performance degrades as the quality of data drops (Figure 4).
* Using DFN for filtering (inference): DFN filters out image-text pairs whose cosine similarities are lower than a threshold (Section 2).

Based on the insight above, the paper curates the High-Quality Image-Text Pairs (HQITP-350M) dataset containing 357 million image-text samples with human-verified captions to train DFNs. In the experiments, the paper shows that CLIP models trained on DFN-induced datasets (1) achieve state-of-the-art performance, (2) improve model efficiency (i.e., training a smaller model that achieves similar performance with previous larger models), and (3) improve VQA performance. The paper will release the DFN-2B dataset filtered by DFN to facilitate future research.

**Strengths:**

1. The paper studies an important problem of filtering training data for CLIP.
2. The paper provides insight into the problem—high-quality pretraining data for DFN matters.
3. The proposed method achieves state-of-the-art performance with better model efficiency.
4. The paper will release the DFN-2B dataset, which is a good contribution to the research community.
5. The paper is well-written and easy to follow.

**Weaknesses:**

### Major Concern

**[Details of Curating HQITP-350M]** The paper does not provide many details about how to create the high-quality HQITP-350M dataset for training DFNs. This is really important because it is the key for training a good DFN as the paper claims. For example, how does the human verification process work? For example, do human annotators make a binary decision of whether image and text are matched or give a score on the scale of 0 to 1? If it is the latter one, how to threshold the scores? How many annotators are requested? What is the source dataset for curating the HQITP-350M dataset? How many image-text pairs are filtered out by humans?

### Minor Concern

**[Definition of Dataset Quality]** The paper defines the “quality” of CLIP’s training data as the image-text similarity. I wonder if the authors consider other aspects to define the data quality, such as the quality of images and the quality of captions [1].

### Minor suggestions

For evaluating the robustness against distributional shifts, the paper can also discuss or include more evaluation on newer OOD ImageNet benchmarks: ImageNet-D, ImageNet-X, and ImageNet-W.

Missing citations to OOD ImageNet datasets used in the paper: ImageNet-Sketch [5],  ImageNet-A [6], ImageNet-V2 [7], ObjectNet [8], ImageNet-R [9] (caption in Figure 5).

### References

[1] Thao Nguyen, Samir Yitzhak Gadre, Gabriel Ilharco, Sewoong Oh, and Ludwig Schmidt, “Improving Multimodal Datasets with Image Captioning,” in NeurIPS Dataset and Benchmark Track, 2023.

[2] Evgenia Rusak, Steffen Schneider, George Pachitariu, Luisa Eck, Peter Vincent Gehler, Oliver Bringmann, Wieland Brendel, and Matthias Bethge, “If your data distribution shifts, use self-learning,” TMLR, 2022.

[3] Badr Youbi Idrissi, Diane Bouchacourt, Randall Balestriero, Ivan Evtimov, Caner Hazirbas, Nicolas Ballas, Pascal Vincent, Michal Drozdzal, et al., “ImageNet-X: Understanding Model Mistakes with Factor of Variation Annotations,” in ICLR, 2023.

[4] Zhiheng Li, Ivan Evtimov, Albert Gordo, Caner Hazirbas, Tal Hassner, Cristian Canton Ferrer, Chenliang Xu, and Mark Ibrahim, “A Whac-A-Mole Dilemma: Shortcuts Come in Multiples Where Mitigating One Amplifies Others,” in CVPR, 2023.

[5] Haohan Wang, Songwei Ge, Eric P. Xing, and Zachary C. Lipton, “Learning Robust Global Representations by Penalizing Local Predictive Power,” in NeurIPS, 2019.

[6] Dan Hendrycks, Kevin Zhao, Steven Basart, Jacob Steinhardt, and Dawn Song, “Natural Adversarial Examples,” in CVPR, 2021.

[7] Benjamin Recht, Rebecca Roelofs, Ludwig Schmidt, and Vaishaal Shankar, “Do ImageNet Classifiers Generalize to ImageNet?,” in ICML, 2019.

[8] Andrei Barbu, David Mayo, Julian Alverio, William Luo, Christopher Wang, Dan Gutfreund, Josh Tenenbaum, and Boris Katz, “ObjectNet: A large-scale bias-controlled dataset for pushing the limits of object recognition models,” in NeurIPS, 2019.

[9] Dan Hendrycks, Steven Basart, Norman Mu, Saurav Kadavath, Frank Wang, Evan Dorundo, Rahul Desai, Tyler Zhu, et al., “The Many Faces of Robustness: A Critical Analysis of Out-of-Distribution Generalization,” in ICCV, 2021.

**Questions:**

In the rebuttal, I expect the authors to address my concerns regarding the following:
1. Details of collecting the HQITP-350M dataset.
2. More discussion on dataset quality

---

> ### Author Response · Authors · 2023-11-17
> **Response to reviewer Q5jf**
>
> Thank you for your kind review. We hope to address your remaining questions.
>
> To address your questions and concerns about HQITP we provide a copy of our general response from above:
>
> We provide some clarity about the HQITP-350M dataset and what we omitted from the paper.
> 1. While we cannot release HQITP-350M for legal purposes, we have shown it is possible to build a DFN better than the existing OAI-CLIP models using only publicly available data. While we only did this up to the “large” DataComp scale in the original submission, we extend this to the XL scale below and compare it to the previous state of the art filtering network (OAI-CLIP-B/32). Our DFN with none of the additional modifications (fine-tuning/open-ai-init/augmentation) from section 4 in the paper outperforms the OAI DFN.
> 2. Because  HQITP-350M dataset would be similar in nature to licensed stock image datasets, so we don’t have access to the exact labeling decision behind each image caption. We also note we will release the dataset induced by our best performing DFN which is sufficient for the community to train strong models.
> 3. In table 2 (above), We provide an additional ablation below where we control the number of examples from HQITP used to train the DFN and measure its performance on the datacomp benchmark (at the medium scale). We show that one can achieve most of the gains with a fraction of the of the full dataset. We note this experiment was done with an earlier version of HQITP that only contained 135M examples.
>
>
> We use image-text similarity as a proxy for quality, but it is flawed and is not truly measuring quality. We attempted to do something similar for images with a binary filter, as well as image and text separately with a M3AE model, but found that empirically CLIP similarity score works best. See Table 1 and Section 3.3 for additional details. Nguyen et al., changes the captions, and can be combined with the methods presented in our work; however, we did not do so because generating new captions can be quite expensive, especially when compared to calculating CLIP scores.
>
> Thank you for your additional suggestions - we have added the citations and will evaluate the additional shifts in a future revision.

---

> > ### Comment · Reviewer_Q5jf · 2023-11-22
> >
> > I appreciate the authors' response. I have read other reviewers' comments. I keep my rating as "8: accept, good paper."

---

### Official Review · Reviewer_NYkb · 2023-11-02

**Soundness:** 4 excellent
**Presentation:** 4 excellent
**Contribution:** 4 excellent
**Rating:** 8
**Confidence:** 3

**Summary:**

The paper compares the performance of (primarily) various configurations of CLIP models to filter image-text datasets for subsequently pretraining vision transformers. The authors make interesting empirical observations regarding the same, and are able to achieve an impressive accuracy vs. compute tradeoff on image-net using their best configuration of data filtering networks.

**Strengths:**

- SoTA compute vs. accuracy trade-off on image-net.
- Thorough & insightful analyses / recipes to understand the effect of different kinds of confounders while training data-filtering networks on downstream performance.
- Expected public release of the best-performing filtered datasets.

**Weaknesses:**

- The proposed approach is still a heuristic for filtering data, and is limited to pointwise filtering for the sake of parallelizability.
- The cost of filtering: (1) training the data filtering networks; and (2) linear inference over the entire dataset; might be too much to be amortized or ignored in the final compute vs. accuracy trade-off (more in questions).

**Questions:**

- Can you highlight the cost of filtering in e.g. Figure 1?
- (Not relevant to the final rating) What do you think about applying these data filtering networks to other domains, e.g., text, or speech, or music?

---

> ### Author Response · Authors · 2023-11-17
> **Response to reviewer NYkb**
>
> Thank you for your kind review. We hope to address your remaining questions.
>
> We agree that pointwise filtering is limited, but is computationally important due to parallelizability; however, this approach can be combined with other dataset filtering approaches, such as restricting keyword counts in the text (e.g. MetaCLIP).
>
> While the cost of filtering is not reflected in Figure 1, DFN, DC-1B, and LAION all rely on CLIP filtering to filter the dataset. DFN and LAION use a ViT B/32 model, which is cheaper (both training and inference) than the ViT-L/14 used in DC-1B. Additionally, the DFN we train for filtering is trained for less samples seen than the OpenAI models used to filter DC-1B and LAION (5B vs 12.8B). Finally, the cost of training a ViT-B/32 (4.4 GMAC at 224px) is much smaller than that of training the best models, such as a ViT-L/14 (81.1 GMAC at 224px) and a ViT-H/14 (167.4 GMAC at 224px). So the total cost of training our best filtering model is 22e9 GMACs, while the total cost of training our ViT-L and ViT-H models are 1e12 GMACs and  6.5e12 GMACS respectively. This corresponds to a filtering overhead of 2% and 0.3% respectively. Unfortunately, we are unable to directly compare with approaches used in proprietary datasets like OAI-WIT-400M and WebLI, as their creation processes are not public knowledge.
>
> We do plan on applying similar methods to text, as well as additional multimodal datasets, and are excited to further explore this space in dataset creation.

---

> > ### Comment · Reviewer_NYkb · 2023-11-23
> > **Response to reviewers**
> >
> > Thank you for your response. I would like to stick to my original rating.

---

### Author Response · Authors · 2023-11-17
**Overall comments**

We thank the reviewers for their kind and helpful reviews. We have uploaded a new revision that reflects changes made in response to the suggestions we have received. We address overall concerns here and go into details in particular review comments below. In order to provide more context to all the reviewers, we have added comparisons to external models trained with similar compute budgets (SiGLIP and MetaCLIP) that were released recently. Our model (when fine-tuned at a comparable resolution) out-performs these models on ImageNet, VTAB, and average metrics on the DataComp evaluation benchmark. We have updated the manuscript with these results and provide a detailed results table below in Tables 3 and 4.

Multiple reviewers have brought up the lack of detail about HQITP-350M and we provide some clarity about the dataset and what we omitted from the paper.
1. While we cannot release HQITP-350M for legal purposes, we have shown it is possible to build a DFN better than the existing OAI-CLIP models using only publicly available data. While we only did this up to the “large” DataComp scale in the original submission, we extend this to the XL scale below and compare it to the previous state of the art filtering network (OAI-CLIP-B/32). Our DFN with none of the additional modifications (fine-tuning/open-ai-init/augmentation) from section 4 in the paper, outperforms the OAI DFN.
2. Because  HQITP-350M dataset would be similar in nature to licensed stock image datasets, so we don’t have access to the exact labeling decision behind each image caption. We also note we will release the dataset induced by our best performing DFN which is sufficient for the community to train strong models.
3. We provide an additional ablation below where we control the number of examples from HQITP used to train the DFN and measure its performance on the datacomp benchmark (at the medium scale). We show that one can achieve most of the gains with a fraction of the of the full dataset. We note this experiment was done with an earlier version of HQITP that only contained 135M examples.

---

> ### Author Response · Authors · 2023-11-17
> **Table 1 + 2**
>
> Table 1
> | DFN             | Model    | Samples Seen | Average | ImageNet 1k | Caltech-101 | CIFAR-10 | CIFAR-100 | CLEVR Counts | CLEVR Distance | Country211 | Describable Textures | EuroSAT | FGVC Aircraft | Food-101 | GTSRB | ImageNet Sketch | ImageNet v2 | ImageNet-A | ImageNet-O | ImageNet-R | KITTI Vehicle Distance | MNIST | ObjectNet | Oxford Flowers-102 | Oxford-IIIT Pet | Pascal VOC 2007 | PatchCamelyon | Rendered SST2 | RESISC45 | Stanford Cars | STL-10 | SUN397 | SVHN  | Flickr | MSCOCO | WinoGAViL | iWildCam | Camelyon17 | FMoW  | Dollar Street | GeoDE |
> | --------------- | -------- | ------------ | ------- | ----------- | ----------- | -------- | --------- | ------------ | -------------- | ---------- | -------------------- | ------- | ------------- | -------- | ----- | --------------- | ----------- | ---------- | ---------- | ---------- | ---------------------- | ----- | --------- | ------------------ | --------------- | --------------- | ------------- | ------------- | -------- | ------------- | ------ | ------ | ----- | ------ | ------ | --------- | -------- | ---------- | ----- | ------------- | ----- |
> | Public-Data-DFN | Vit-L/14 | 12.8B        | 0.658   | 0.770       | 0.948       | 0.980    | 0.872     | 0.395        | 0.204          | 0.261      | 0.655                | 0.690   | 0.387         | 0.932    | 0.619 | 0.662           | 0.696       | 0.591      | 0.386      | 0.897      | 0.395                  | 0.921 | 0.704     | 0.815              | 0.935           | 0.815           | 0.602         | 0.505         | 0.677    | 0.945         | 0.990  | 0.736  | 0.694 | 0.815  | 0.549  | 0.509     | 0.144    | 0.579      | 0.145 | 0.661         | 0.911 |
> | OAI-CLIP-B/32   | Vit-L/14 | 12.8B        | 0.636   | 0.755       | 0.935       | 0.980    | 0.859     | 0.382        | 0.249          | 0.250      | 0.619                | 0.738   | 0.287         | 0.920    | 0.602 | 0.635           | 0.685       | 0.575      | 0.373      | 0.871      | 0.138                  | 0.774 | 0.685     | 0.744              | 0.931           | 0.811           | 0.507         | 0.599         | 0.707    | 0.891         | 0.990  | 0.733  | 0.573 | 0.813  | 0.551  | 0.497     | 0.142    | 0.567      | 0.234 | 0.660         | 0.912 |
>
>
>
> Table 2
>
> | Percent of HQITP-135M | ImageNet | ImageNet dist. shifts | VTAB  | Retrieval | Average full |
> | --------------------- | -------- | --------------------- | ----- | --------- | ------------ |
> | 10 percent            | 0.295    | 0.237                 | 0.341 | 0.273     | 0.333        |
> | 20 percent            | 0.312    | 0.253                 | 0.371 | 0.277     | 0.352        |
> | 30 percent            | 0.313    | 0.256                 | 0.354 | 0.284     | 0.347        |
> | 40 percent            | 0.320    | 0.259                 | 0.371 | 0.284     | 0.352        |
> | 50 percent            | 0.324    | 0.258                 | 0.350 | 0.282     | 0.340        |
> | 60 percent            | 0.321    | 0.261                 | 0.361 | 0.285     | 0.350        |
> | 70 percent            | 0.323    | 0.259                 | 0.357 | 0.282     | 0.350        |
> | 80 percent            | 0.326    | 0.262                 | 0.366 | 0.284     | 0.353        |
> | 90 percent            | 0.322    | 0.260                 | 0.369 | 0.288     | 0.351        |
> | 100 percent           | 0.327    | 0.266                 | 0.372 | 0.270     | 0.353        |

---

> > ### Author Response · Authors · 2023-11-17
> > **Table 3**
> >
> > ## Table 3 (DFN-2B experiments)
> >
> > | Dataset            | Model    | Samples Seen | Average | ImageNet 1k | Caltech-101 | CIFAR-10 | CIFAR-100 | CLEVR Counts | CLEVR Distance | Country211 | Describable Textures | EuroSAT | FGVC Aircraft | Food-101 | GTSRB  | ImageNet Sketch | ImageNet v2 | ImageNet-A | ImageNet-O | ImageNet-R | KITTI Vehicle Distance | MNIST  | ObjectNet | Oxford Flowers-102 | Oxford-IIIT Pet | Pascal VOC 2007 | PatchCamelyon | Rendered SST2 | RESISC45 | Stanford Cars | STL-10 | SUN397 | SVHN   | Flickr | MSCOCO | WinoGAViL | iWildCam | Camelyon17 | FMoW   | Dollar Street | GeoDE  |
> > | ------------------ | -------- | ------------ | ------- | ----------- | ----------- | -------- | --------- | ------------ | -------------- | ---------- | -------------------- | ------- | ------------- | -------- | ------ | --------------- | ----------- | ---------- | ---------- | ---------- | ---------------------- | ------ | --------- | ------------------ | --------------- | --------------- | ------------- | ------------- | -------- | ------------- | ------ | ------ | ------ | ------ | ------ | --------- | -------- | ---------- | ------ | ------------- | ------ |
> > | DFN-2B-No-Finetune | ViT-L/14 | 12.8B        | 0.666   | 0.786       | 0.947       | 0.985    | 0.880     | 0.303        | 0.247          | 0.315      | 0.685                | 0.624   | 0.421         | 0.942    | 0.586  | 0.679           | 0.717       | 0.640      | 0.357      | 0.899      | 0.204                  | 0.910  | 0.726     | 0.873              | 0.949           | 0.831           | 0.649         | 0.593         | 0.734    | 0.948         | 0.985  | 0.755  | 0.663  | 0.819  | 0.555  | 0.519     | 0.158    | 0.655      | 0.195  | 0.666         | 0.918  |
> > | DFN-2B             | ViT-L/14 | 12.8B        | 0.669   | 0.814       | 0.953       | 0.984    | 0.884     | 0.334        | 0.249          | 0.282      | 0.661                | 0.646   | 0.396         | 0.946    | 0.616  | 0.683           | 0.745       | 0.668      | 0.392      | 0.900      | 0.201                  | 0.847  | 0.739     | 0.866              | 0.955           | 0.816           | 0.630         | 0.551         | 0.733    | 0.947         | 0.977  | 0.755  | 0.654  | 0.824  | 0.570  | 0.552     | 0.189    | 0.626      | 0.222  | 0.688         | 0.910  |
> > | DFN-2B             | ViT-L/14 | 39B          | 0.678   | 0.822       | 0.950       | 0.986    | 0.893     | 0.340        | 0.232          | 0.320      | 0.668                | 0.682   | 0.483         | 0.950    | 0.633  | 0.704           | 0.757       | 0.675      | 0.361      | 0.918      | 0.239                  | 0.875  | 0.657     | 0.878              | 0.961           | 0.847           | 0.642         | 0.582         | 0.730    | 0.947         | 0.989  | 0.759  | 0.657  | 0.847  | 0.596  | 0.555     | 0.186    | 0.654      | 0.182  | 0.682         | 0.925  |
> > | LAION-2B           | ViT-L/14 | 34B          | 0.621   | 0.753       | 0.939       | 0.966    | 0.833     | 0.312        | 0.223          | 0.263      | 0.629                | 0.646   | 0.365         | 0.910    | 0.562  | 0.633           | 0.678       | 0.539      | 0.387      | 0.874      | 0.229                  | 0.541  | 0.653     | 0.748              | 0.931           | 0.805           | 0.564         | 0.593         | 0.669    | 0.926         | 0.989  | 0.743  | 0.409  | 0.825  | 0.549  | 0.439     | 0.126    | 0.597      | 0.201  | 0.640         | 0.892  |
> > | DFN-2B-No-Finetune | ViT-B/16 | 12.8B        | 0.612   | 0.7288      | 0.9351      | 0.968    | 0.8228    | 0.199        | 0.2461         | 0.22       | 0.609                | 0.4374  | 0.2721        | 0.9059   | 0.5413 | 0.6054          | 0.6497      | 0.4404     | 0.4525     | 0.8296     | 0.3432                 | 0.8462 | 0.6154    | 0.8207             | 0.9243          | 0.794           | 0.4982        | 0.5585        | 0.6425   | 0.9115        | 0.9736 | 0.7059 | 0.6626 | 0.7664 | 0.4996 | 0.5247    | 0.1257   | 0.4973     | 0.1492 | 0.625         | 0.8998 |
> > | DFN-2B             | ViT-B/16 | 12.8B        | 0.609   | 0.7624      | 0.9429      | 0.9672   | 0.8347    | 0.2323       | 0.2453         | 0.1955     | 0.5755               | 0.54    | 0.2485        | 0.913    | 0.4699 | 0.6207          | 0.682       | 0.4821     | 0.493      | 0.831      | 0.1927                 | 0.782  | 0.6319    | 0.8199             | 0.9369          | 0.7885          | 0.5215        | 0.4865        | 0.6138   | 0.9073        | 0.9753 | 0.7142 | 0.599  | 0.7728 | 0.5188 | 0.5417    | 0.1556   | 0.4993     | 0.1411 | 0.625         | 0.891  |

---

> > > ### Author Response · Authors · 2023-11-17
> > > **Table 4**
> > >
> > > ## Table 4 (DFN-5B) experiments
> > >
> > > | Dataset  | Model                    | Samples Seen | Average | ImageNet 1k | Caltech-101 | CIFAR-10 | CIFAR-100 | CLEVR Counts | CLEVR Distance | Country211 | Describable Textures | EuroSAT | FGVC Aircraft | Food-101 | GTSRB  | ImageNet Sketch | ImageNet v2 | ImageNet-A | ImageNet-O | ImageNet-R | KITTI Vehicle Distance | MNIST  | ObjectNet | Oxford Flowers-102 | Oxford-IIIT Pet | Pascal VOC 2007 | PatchCamelyon | Rendered SST2 | RESISC45 | Stanford Cars | STL-10 | SUN397 | SVHN  | Flickr | MSCOCO | WinoGAViL | iWildCam | Camelyon17 | FMoW   | Dollar Street | GeoDE  |
> > > | -------- | ------------------------ | ------------ | ------- | ----------- | ----------- | -------- | --------- | ------------ | -------------- | ---------- | -------------------- | ------- | ------------- | -------- | ------ | --------------- | ----------- | ---------- | ---------- | ---------- | ---------------------- | ------ | --------- | ------------------ | --------------- | --------------- | ------------- | ------------- | -------- | ------------- | ------ | ------ | ----- | ------ | ------ | --------- | -------- | ---------- | ------ | ------------- | ------ |
> > > | DFN-5B   | ViT-H/14                 | 39B          | 0.698   | 0.834       | 0.955       | 0.988    | 0.905     | 0.297        | 0.212          | 0.344      | 0.706                | 0.655   | 0.715         | 0.957    | 0.677  | 0.727           | 0.774       | 0.699      | 0.381      | 0.930      | 0.335                  | 0.858  | 0.765     | 0.900              | 0.966           | 0.818           | 0.654         | 0.546         | 0.751    | 0.958         | 0.989  | 0.769  | 0.676 | 0.865  | 0.631  | 0.556     | 0.203    | 0.705      | 0.208  | 0.701         | 0.929  |
> > > | DFN-5B   | ViT-H/14-378px           | 44B          | 0.710   | 0.844       | 0.952       | 0.988    | 0.904     | 0.360        | 0.209          | 0.379      | 0.711                | 0.613   | 0.722         | 0.962    | 0.678  | 0.732           | 0.783       | 0.796      | 0.381      | 0.938      | 0.397                  | 0.836  | 0.797     | 0.894              | 0.970           | 0.824           | 0.696         | 0.555         | 0.759    | 0.960         | 0.991  | 0.773  | 0.674 | 0.880  | 0.638  | 0.567     | 0.221    | 0.721      | 0.208  | 0.717         | 0.935  |
> > > | Webli    | ViT-SO400M-14-SigLIP-384 | 45B          | 0.694   | 0.831       | 0.960       | 0.967    | 0.836     | 0.407        | 0.225          | 0.365      | 0.730                | 0.635   | 0.607         | 0.964    | 0.643  | 0.745           | 0.772       | 0.825      | 0.278      | 0.958      | 0.208                  | 0.886  | 0.827     | 0.911              | 0.968           | 0.717           | 0.527         | 0.700         | 0.721    | 0.952         | 0.993  | 0.754  | 0.515 | 0.886  | 0.633  | 0.575     | 0.229    | 0.615      | 0.331  | 0.730         | 0.933  |
> > > | MetaCLIP | ViT-H/14                 | 12.8B        | 0.667   | 0.8051      | 0.9536      | 0.9804   | 0.8634    | 0.2115       | 0.1881         | 0.3716     | 0.7271               | 0.645   | 0.5114        | 0.9423   | 0.6257 | 0.7052          | 0.7417      | 0.7533     | 0.304      | 0.9342     | 0.2771                 | 0.7266 | 0.7642    | 0.8448             | 0.9561          | 0.7495          | 0.6222        | 0.6925        | 0.7024   | 0.899         | 0.9944 | 0.744  | 0.591 | 0.8507 | 0.5752 | 0.5312    | 0.168    | 0.5782     | 0.2314 | 0.6811        | 0.9077 |

---

### Meta-Review · Area_Chair_znym · 2023-12-06

**Metareview:**

The paper focuses on the problem of dataset filtering for contrastive language-image pre-training (CLIP). It introduces data filtering networks (DFN), which are trained to filter image-text pairs based on their similarity embeddings. The key observation is that the accuracy of a DFN on a given task does not necessarily predict how well it selects a subset of data for that task. The paper proposes training DFNs on high-quality image-caption datasets and fine-tuning them on important datasets. The experiments show that this approach leads to competitive performance on various tasks, including ImageNet Zero-Shot, ImageNet Distribution Shift, VTAB, and Retrieval, when filtering different data sizes.

Strengths:
* The exploration of DFN and their impact on dataset selection for CLIP.
* The interesting observation that DFN accuracy does not necessarily correlate with the quality of data selected for downstream tasks.
* The competitive performance achieved on various tasks, such as ImageNet Zero-Shot, ImageNet Distribution Shift, VTAB, and Retrieval, when using the proposed DFNs.
* The paper's clear and well-written presentation, making it easy to follow.

Weaknesses:
* The heavy reliance on fine-tuning for performance improvements, particularly on specific datasets like MS COCO and Flickr30k, which may limit the generalization of the proposed approach to other downstream tasks.
* Discussion on dataset quality beyond image-text similarity could be expanded. This may be important for understanding the effectiveness of data filtering.
* Potential questions regarding the trade-offs in accuracy on specific datasets when using different DFNs for filtering.

**Justification For Why Not Higher Score:**

Weaknesses include the heavy reliance on fine-tuning for performance improvements, which may limit generalization.

**Justification For Why Not Lower Score:**

Strengths of the paper include its exploration of DFNs, the observation regarding the lack of correlation between DFN accuracy and data selection quality, and the competitive performance achieved on several tasks.

---

### Decision · Program_Chairs · 2024-01-16

Accept (poster)